# Hypergraph-based Zero-shot Multi-modal Product Attribute Value Extraction

## Abstract

It is essential for e-commerce platforms to provide accurate, complete, and timely product attribute values, in order to improve the search and recommendation experience for both customers and sellers. In the real-world scenario, it is difficult for these platforms to identify attribute values for the newly introduced products given no similar product history records for training or retrieval. Besides, how to jointly learn the product representation given various product information in multiple modalities, such as textual modality (e.g., product titles and descriptions) and visual modality (e.g., product images), is also a challenging task. To address these limitations, we propose a novel method for extracting multi-label product attribute-value pairs from multiple modalities in the zero-shot scenario, where labeled data is absent during training. Specifically, our method constructs heterogeneous hypergraphs, where product information from different modalities is represented by different types of nodes, and the text and image nodes are embedded and learned through CLIP encoders to effectively capture and integrate multi-modal product information. Then, the complex interrelations among these nodes are modeled through the hyperedges. By learning informative node representations, our method can accurately predict links between unseen product nodes and attribute-value nodes, enabling zero-shot attribute value extraction. We conduct extensive experiments and ablation studies on several categories of the public MAVE dataset and the results demonstrate that our proposed method significantly outperforms several state-of-the-art generative model baselines in multi-label, multi-modal product attribute value extraction in the zero-shot setting.

## Keywords

Attribute value extraction, Multi-modal learning, Zero-shot learning, Heterogeneous hypergraph

## 1 Introduction

Product attribute values are crucial in e-commerce as they provide valuable product information that enables customers to search, compare, and purchase their desired products more effectively and efficiently. Moreover, these values assist sellers in accurately representing and categorizing their products [59]. However, manual labeling by sellers often leads to incomplete or mislabeled attribute values, which could impede the searching and purchasing process for customers. To address these challenges, product attribute value extraction (AVE), which aims to automatically extract attribute values from product information, such as product titles, descriptions, and images, is an important task in the e-commerce field [59].

Most existing approaches for AVE rely on traditional supervised learning models, which require large quantities of labeled training data. However, manually labeling new products is often time-consuming and labor-intensive [16]. Consequently, it is essential to extract unseen attribute values for new products where labeled

training data is unavailable (i.e., in the zero-shot setting). In addition, many existing works focus on extracting attribute values from solely textual information, such as product titles and descriptions [39, 45, 46]. In real life, substantial attribute value information is implicitly contained within visual modality (i.e., product images), which is often overlooked compared to more explicit textual information [40, 52]. For instance, Figure 1 gives an example of the product data in our dataset, which includes both textual information, such as the product title and description, as well as visual information in the form of a product image. Moreover, though existing

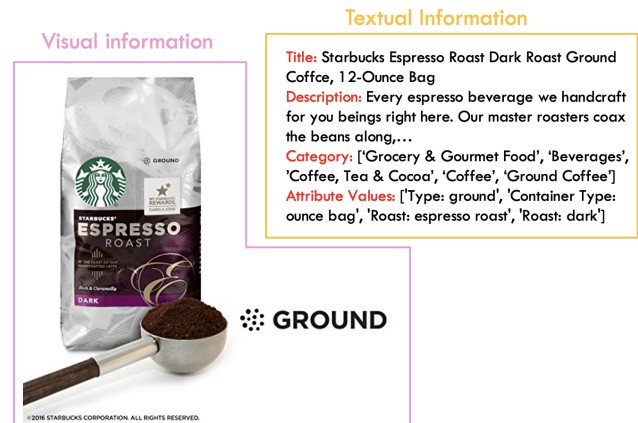

**Figure 1: An example of a product profile, including both textual and visual information.**

SOTA generative multi-modal large language models (MLLMs) (i.e. LLaVA [28], Qwen [1], etc.) can handle multi-modal data input for a zero-shot attribute value prediction, these approaches are typically limited to generating a single attribute value at a time, lacking the capability to simultaneously extract multiple attribute-value pairs. This limitation is impractical for real-world applications, where products often possess several attribute-value pairs. Efficient extraction of these attribute values simultaneously is crucial for saving computational resources and reducing computational time.

To tackle the above challenges, we propose a hypergraph-based model for multi-modal, multi-label attribute value extraction in the zero-shot setting. Our model employs heterogeneous hypergraphs to capture intricate, higher-order interrelations among various types of nodes. These interrelations are derived from different aspects of product information or are influenced by user behaviors. Subsequently, the learned hypergraphs are capable of inferring connections between unseen product nodes and attribute value nodes. Specifically, we introduce four types of nodes, each representing products (i.e., product titles and descriptions), product categories, product attribute values, and product images. We first construct graphs to capture initial correlations and interactions

among these nodes; based on these graphs, we then construct a unified hypergraph to further capture more complex and higher-order interrelations influenced by user behaviors (i.e., 'also view', 'also buy') and product inventory information (e.g., 'product with all images') between nodes. The final node representations learned from the hypergraph encapsulate accurate and expressive information regarding products and product attribute values, demonstrating excellent capabilities in predicting links between new products and unseen attribute values. We conduct extensive experiments across 8 different product categories within the MAVE dataset, and the results indicate that our multi-modal hypergraph-based AVE model outperforms several existing methods in the zero-shot setting from the perspective of prediction performance and computational efficiency (time, GPU usage, and model weight). Furthermore, ablation studies show that integrating the visual modality can significantly improve the effectiveness of our model while introducing the hypergraph module also substantially improves the model performance compared to directly inferring from graphs.

Our main contributions are summarized as follows:

- We propose a zero-shot multi-modal, multi-label model to extract unseen attribute values for new products, which integrates an CLIP encoder to better learn the representations for both textual and visual product input data.
- We construct heterogeneous hypergraphs to capture complex and higher-order correlations and interactions from basic product information and user behavior information.
- Extensive experiments on the public dataset MAVE indicates that our proposed model significantly outperforms (higher F1-score and better computational efficiency) several SOTA generative MLLMs in zero-shot learning.

## 2 Related Work

### 2.1 Attribute Value Extraction

Product attribute value extraction aims at identifying attribute values based on the product information. Taking advantage of pre-trained large language models (LLMs), most existing studies focus on extracting attribute values from product titles or descriptions by using classification models [11, 12, 15], transformers [6], and generative LLMs [2, 14, 24, 34, 35, 37]. Some recent works explore the product visual features to enhance the product attribute value extraction by utilizing multi-modal transformer [40], combining OCR [27], using multi-modal attention mechanism [10, 52], prompt-tuning of pre-trained transformer [47], and directly leveraging multi-modal large language models (MLLMs) [56, 57] to generate product attribute values from the combination of product texts and images. However, these methods utilizing large (multi-modal) language models require large quantities of labeled training data for fine-tuning the pre-trained MLLMs. Even the works directly use the SOTA pre-trained MLLMs (i.e. Qwen [1], LLaVA [28]) for zero-shot attribute value generation still require computing resources for model inference [56] and demonstrate limited multi-label attribute value generation shown in Sec. 5.1. Besides, these works also miss the complex interrelations among different products. For multi-label zero-shot attribute value extraction, the most related work explores using heterogeneous hypergraph to do inductive link

prediction [17]. Nevertheless, it only considers text modality and misses the rich information and correlations from product images.

### 2.2 Multi-modal Zero-shot Learning

Multi-modal zero-shot learning (MZSL) has been widely applied in computer vision and natural language processing [5]. MZSL can be roughly divided into three main categories: (1) embedding-based methods, where the model directly constructs the mapping relationship between visual space and semantic space. One popular approach is the contrastive learning for MZSL [21] (i.e. CLIP [33], SigLIP [51], DreamLIP [54], etc.); (2) generative models, where ZSL is treated as a data-missing problem and representations of data in one modality is generated from another modality. For example, conditional variational autoencoders [31] or generative adversarial networks [18] are used to be trained on seen classes and then infer the characteristics of unseen classes in MZSL; (3) graph-based methods, where relationships between different classes and modalities are captured by graph links [4, 30, 43]. We propose to combine the embedding-based and graph-based approaches to leverage the strengths of both for MZSL on attribute value extraction.

### 2.3 Multi-modal Graph Learning

Although many current multi-modal learning techniques concentrate on mapping or aligning various modalities, real-world data is often more intricate, featuring complex many-to-many relationships that can be effectively modeled using graphs. Multi-modal graph learning (MGL) aims to tackle this complexity by leveraging graph structures with relational representations to fully explore both the inter-modal and intra-modal correlations from multi-modal data among multiple multi-modal neighbors [13, 32, 55]. Recent works in MGL have focused on integrating multi-modal features with graph or hypergraph structures (i.e. MGCN [22, 42], MGAN [23, 53], MGCL [25, 29], MHN [19, 49, 50]) across a wide range of downstream tasks (i.e. knowledge graph completion [36, 38, 41], recommendation [3, 19, 29], disease prediction [23, 44], etc.). Motivated by multi-modal fusion for hypergraphs with contrastive learning [19, 49] and the zero-shot ability from inductive link prediction in graphs [17], we propose a multi-modal hypergraph-based model with text and image nodes embedded through the CLIP encoder for zero-shot attribute value extraction in e-Commerce.

## 3 Methodology

### 3.1 Problem Formulation

In this section, we formally formulate the problem of heterogeneous hypergraph-based multi-modal zero-shot AVE, and provide some annotations for it. *The problem formulation is based on [16].* The product profile consists of product title and description, category, attribute values, and images. We denote a product profile as $D = (P_t, P_v)$, where $P_t$ is the textual modality of the product profile and $P_v$ is the visual modality. For the textual modality, $P_t = (p_t^d, p_t^c, p_t^a)$, where $p_t^d, p_t^c, p_t^a$ represent product title and description, product category, and product attribute value, respectively. For the visual modality, $P_v = (p_v^i)$, where $p_v^i$ represents product image.

Note that each product may have multiple images captured from different angles or perspectives to provide a comprehensive representation of its attributes and corresponding values. As illustrated in Figure 2, the product 'cat bowl' is depicted with three product images, each suggesting distinct attribute value information from different perspectives. For example, the top-left image demonstrates the design details of the cat bowl, the top-right image highlights its capacity, and the bottom image indicates that the bowl is specifically intended for cats. Also, each product may belong to multiple categories and have multiple attribute values. Still taking the cat bowl as an example, it is classified under categories such as Pet Supplies, Cats, and Bowls & Dishes, and has multiple attribute values, including Type: Pet Dish and Pet: Cat.

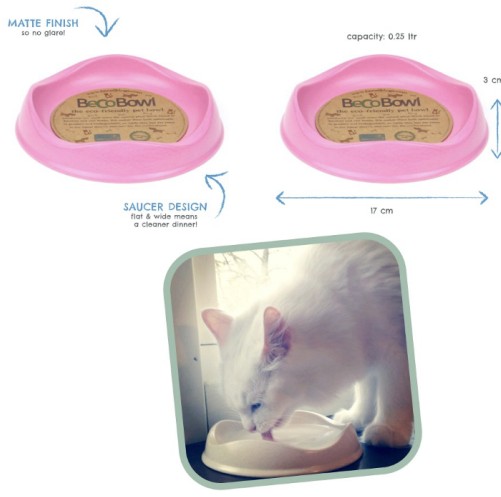

**Figure 2: A product, Cat Bowl, with multiple images.**

Given a product profile $D$, the goal of zero-shot AVE is to determine whether there is a link between a new product $p$ and unseen attribute values $p_t^a$. The likelihood of such a link is calculated by the cosine similarity between $p$ and $p_t^a$.

## 3.2 Model Overview

Figure 3 presents the overall framework of our proposed method. Essentially, our framework is composed of three main components: heterogenous hypergraph construction, hypergraph relation learning, and zero-shot link prediction. The details of each component are as follows:

*3.2.1 Heterogeneous Hypergraph Construction. Product inventory* refers to structured data that provides information about the attributes and characteristics of a product. In our context, product inventory data includes product titles and descriptions, product attribute values, product categories, and product images. *User behavior information* in e-commerce refers to the actions and activities that reflect how customers interact with e-commerce platforms, offering valuable insights into how certain products relate to others. In our case, we focus on two types of user behavior information: *'also view'* and *'also buy'*. The 'also view' behavior means people

who view product A also view product B, and the 'also buy' behavior means people who buy product A also buy product B.

To enable hypergraphs to effectively capture correlations and interactions between product inventory data and user behavior information, we begin by constructing *graphs* that will serve as the basis for hypergraph construction. For graph construction, we define four types of nodes: *Products (P), Categories (C), Attribute Values (A), and Images (I)*. To generate initial node representations, we employ the text encoder from the CLIP model [33] to encode textual information for $P$, $C$, and $A$, while utilizing the CLIP image encoder to encode visual information for $I$. The initial node representations for $P$, $C$, $A$, and $I$ are formulated as follows:

$$h_p = CLIP_{text}(p_t^d) \tag{1}$$

$$h_c = CLIP_{text}(p_t^c) \tag{2}$$

$$h_a = CLIP_{text}(p_t^a) \tag{3}$$

$$h_i = CLIP_{image}(p_v^i) \tag{4}$$

To establish connections between nodes, we introduce three types of edges: *product in category, product has attribute values, and product has images*. We formally define the edge set as:

$$E = \{e^{pc}, e^{pa}, e^{pi}\} \tag{5}$$

where $e^{pc}$ denotes *product in category* edge, $e^{pa}$ denotes *product has attribute values* edge, $e^{pi}$ denotes *product has images* edge. Figure 3 Stage 1 demonstrates the graph construction process. By integrating these three subgraphs, we obtain a unified graph that encapsulates all relevant information from the product inventory.

The bottom part in Figure 3 Stage 1 demonstrates the process of *hypergraph construction* based on the information contained within the unified graph. We introduce two primary types of hyperedges, *also view* and *also buy*, which represent the previously described user behavior information. *also view* and *also buy* hyperedges capture the relationships between products and product images that align with the same user behavior. Additionally, we introduce another type of hyperedge, *product with all images*, which records the relationships between products and their corresponding images. Formally, we define the hyperedge set as:

$$\mathcal{E} = \{\epsilon^{vp}, \epsilon^{vi}, \epsilon^{bp}, \epsilon^{bi}, \epsilon^{pi}\} \tag{6}$$

where $\epsilon^{vp}$ denotes *product also view* hyperedge, $\epsilon^{vi}$ denotes *image also view* hyperedge, $\epsilon^{bp}$ denotes *product also buy* hyperedge, $\epsilon^{bi}$ denotes *image also buy* hyperedge, and $\epsilon^{pi}$ denotes *product with all images* hyperedge. As shown in Figure 3 Stage 1, we use a single *also view* hyperedge to join multiple products that are viewed by the same user, and a single *also buy* hyperedge to join multiple products that are purchased by the same user. Additionally, images associated with the joined products are also connected using hyperedges of the same type. By doing this, we aim to exploit implicit correlated interrelations between products and images that may initially appear unrelated. Furthermore, a single *product with all images* hyperedge is utilized to connect each product with its corresponding images, facilitating the capture of interactions between the textual and visual modalities.

*3.2.2 Hypergraph Relation Learning.* By leveraging the structural information of the heterogeneous hypergraph (as shown in Figure 3 Stage 1), we perform hypergraph relation learning to fully identify

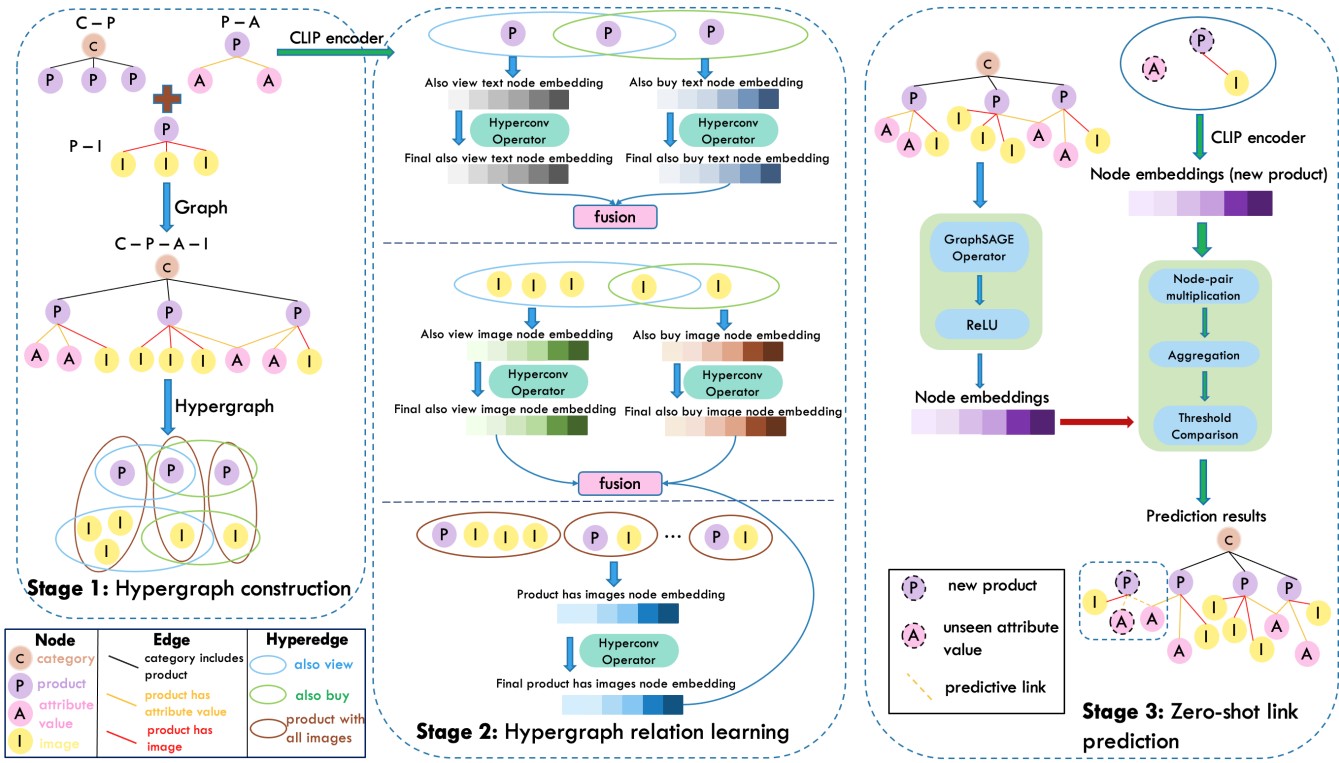

**Figure 3: Overview of our proposed model architecture. The model is composed of three main components: (1) Hypergraph construction: constructs hypergraphs based on product inventory data and user behavior information. (2) Hypergraph relation learning: captures higher-order relationships between product nodes and image nodes. (3) Zero-shot link prediction: infers links between new products and unseen attribute values.**

and understand the relationships between different entities within the heterogeneous hypergraph. Based on the initial node embeddings obtained from the CLIP text and image encoders, we conduct *message passing* within hypergraphs to enable more complex and informative node representations by aggregating information from neighboring nodes. For example, when calculating higher-order node representations for product nodes (as shown in the upper section in Figure 3 Stage 2), our model performs message passing:

$$m_{N(p_v)}^{(l)} = AGGREGATE^{(l)}(h_{p_u}^{(l)}, \forall u \in N(p_v)) \tag{7}$$

$$h_{p_v}^{(l+1)} = UPDATE^{(l)}(h_{p_v}^{(l)}, m_{N(p_v)}^{(l)}) \tag{8}$$

where $h_{p_u}^{(l)}$ represents the embeddings of the neighboring nodes of node $p_v^{(l)}$, $h_{p_v}^{(l)}$ represents the embedding of the current node $p_v^{(l)}$, AGGREGATE is an aggregation function that aggregates the messages from the neighboring nodes, and UPDATE function is used to update the node embedding for node $p_v^{(l)}$ based on its current features and the aggregated messages. Similarly, message passing for image nodes is calculated as follows:

$$m_{N(i_v)}^{(l)} = AGGREGATE^{(l)}(h_{i_u}^{(l)}, \forall u \in N(i_v)) \tag{9}$$

$$h_{i_v}^{(l+1)} = UPDATE^{(l)}(h_{i_v}^{(l)}, m_{N(i_v)}^{(l)}) \tag{10}$$

where $h_{i_u}^{(l)}$ represents the embeddings of the neighborhood nodes of image node $i_v^{(l)}$, $h_{i_v}^{(l)}$ represents the embedding of the current image node $i_v^{(l)}$. We use the same AGGREGATE function and UPADTE function as previously defined.

When nodes belong to different modalities (i.e., both $h_{p_v}^{(l)}$ and $h_{i_v}^{(l)}$) within the hypergraph (as shown in the lower section of Figure 3 Stage 2), message passing is performed in the same manner. Message passing between nodes of different modalities enables the initial fusion of textual and visual features.

*Multi-modal Fusion.* After obtaining individual node embeddings for the five different types of hyperedges, we fuse the product node embeddings to generate a final unified node representation of products, and similarly, fuse the image node embeddings to create a final unified node representation for images. Moreover, different types of hyperedges may contribute unequally to the final node representations [16]. Therefore, we employ a fusion approach to further fuse node embeddings. This fusion approach considers varying levels of contribution from each hyperedge type, thereby optimizing both the final node representations and overall model performance. For *product also view* and *product also buy* hyperedges, we conduct fusion as follows:

$$h_{p_v} = \alpha \cdot h_{p_v}^{vp} + (1 - \alpha) \cdot h_{p_v}^{bp} \tag{11}$$

where $h_{p_v}^{vp}$ is *product also view* node embedding, $h_{p_v}^{bp}$ is *product also buy* node embedding, $\alpha$ is the weight of *product also view* node embedding, $1 - \alpha$ is the weight of *product also buy* node embedding, and $h_{p_v}$ is the final unified product embedding.

Similarly, for *image also view, image also buy,* and *product with all images* hyperedges, the fusion is calculated as:

$$h_{i_v} = \gamma \cdot h_{i_v}^{vi} + \delta \cdot h_{i_v}^{bi} + (1 - \gamma - \delta) \cdot h_{i_v}^{pi} \tag{12}$$

where $h_{i_v}^{vi}$ is *image also view* node embedding, $h_{i_v}^{bi}$ is *image also buy* node embedding, $h_{i_v}^{pi}$ is *product with all images* node embedding, $\gamma$, $\delta, 1 - \gamma - \delta$ are weights of *image also view, image also buy, product with all images*, respectively, and $h_{i_v}$ is the final unified image embedding. Through the above fusion process, we obtain final product and image embeddings that capture more sophisticated correlations within user behaviors and product inventory information across different types of nodes and hyperedges.

*3.2.3 Zero-shot Link Prediction.* The relation learning stage allows us to obtain higher-level node embeddings that capture more intricate correlations among user behaviors and product inventory for both products and images. Additionally, we incorporate category and attribute value nodes, which are encoded by the CLIP text encoder, into the final graph structure to support the task of zero-shot link prediction. To generate the final node embeddings, all nodes in the final graph are processed through $L$ layers of GraphSAGE [20], followed by the ReLU activation function.

To predict whether there is a link between a new product $p$ and an attribute-value pair $a$, we calculate the cosine similarity between their embeddings to evaluate the probability that product $p$ will have the attribute value $a$. Specifically, for a given product $p_i$ and attribute value $a_i$, the cosine similarity is calculated as follows:

$$S_c(\tilde{h_{p_i}}, \tilde{h_{a_i}}) = \frac{\tilde{h_{p_i}} \cdot \tilde{h_{a_i}}}{\|\tilde{h_{p_i}}\| \cdot \|\tilde{h_{a_i}}\|} \tag{13}$$

where $\tilde{h_{p_i}}$ is the node embedding for the new product $p_i$ and $\tilde{h_{a_i}}$ is the node embedding for the unseen attribute-value pair $a_i$. As shown in Figure 3 Stage 3, we predict whether a link exists between the new product (in a dotted line) and the unseen attribute-value pair (also in a dotted line) based on the previously learned graph. We can also predict links between the new product and existing attribute-value pairs. We employ a binary cross-entropy loss to optimize the performance of our model:

$$L = \sum_{p_i \in P, a_j \in A} y^{p_i a_j} \cdot log(S_c^{p_i a_j}) + (1 - y^{p_i a_j}) \cdot (1 - log(S_c^{p_i a_j j})) \tag{14}$$

where $y^{p_i a_j}$ is the ground truth for product-attribute sample $p_i a_j$, $S_c^{p_i a_j}$ is the predicted probability of the same sample. In addition, we employ the strategy of negative sampling to help train the model.

## 4 Experiments

### 4.1 Dataset

We conduct extensive experiments on eight main categories (Industrial, Arts, Cellphones, Automotive, Office products, Pet, Grocery, and Tools) from the MAVE dataset [48], which which is a large e-Commerce dataset derived from Amazon Review Dataset. For each category, we concatenate the 'title' and 'description' into a single column, which is subsequently used as product node features. We leverage the 'category_a' field, which records multiple subcategories for each product. We clean raw attribute-value pairs data by eliminating duplicate values and erroneous values. We use one or more product images associated with each product as visual information and exclude products with no corresponding images.

To simulate the zero-shot scenario, we split the dataset into training, validation, and test sets, ensuring that no overlap exists between them, following the multi-label zero-shot sampling in [16]. This step is critical since it prevents data leakage between the training, validation, and test stages, thereby guaranteeing a rigorous zero-shot link prediction setting. Note that each time the model is trained, we randomly re-split the data into training, validation, and test sets. Accordingly, we report the overall data statistic in Table 1.

### 4.2 Baselines Setup

Our model is compared with four state-of-the-art (SOTA) generative MLLMs (Qwen-VL [1], LLaVA [28], BLIP2 [26], InternVL [7]) for implicit attribute value extraction [58] in the zero-shot setting. Detailed descriptions of baseline models are in Appendix A.

For each baseline model, we use a standardized prompt template structured as follows: *"What is the <attribute names> of this product: <'product titles and description'>? Answer with the option from the given choices directly: <attribute values>. Answer:"* For example, for a product in the *Pet* category, the prompt would be: *"What is the Material of this product: [Title] All Glass Aquarium AAG1055 Tank, 55-Gallon? Answer with the option from the given choices directly: acrylic, glass, pine, plastic, resin, steel, wood. Answer:".* For products with multiple attribute values, the generative baselines are implemented multiple times (one attribute each time). Providing a standardized prompt format ensures a fair comparison across all baseline models. In addition, we keep only those attributes that have six or more corresponding attribute values for baselines. We randomly sample 2k data points to evaluate the performance of each baseline model as the inference time for baselines is very long shown in Table 4. More details are discussed in Section 6.

### 4.3 Implementation

*Implementation Details.* We implement our model using PyTorch and train it on a single NVIDIA A100 GPU. During training, we optimize the model using the AdamW optimizer. We perform a grid search to identify the optimal hyperparameters, setting the batch size to 256, learning rate to $5e^{-5}$, weight decay to $1e^{-6}$, dropout rate to 0.5, and training epochs to 500, which lead to the best model performance. The hidden size for both the hypergraph convolutional operator and the GraphSAGE operator is set to 512. When sampling neighbors in hypergraphs, we sample 20 neighbors for each node and set the rate of negative sampling to 2. We also limit the maximum length of text sequences to 77, truncating any text sequences that exceed this maximum length. For our model and all ablation studies, we run 10 times using different random seeds. In addition, after evaluating the effects of different weight configurations on model performance, we set the value of $\alpha$ to 0.5, and both $\gamma$ and $\delta$ to 0.3 to optimize the model's performance.

**Table 1: Data statistics of eight categories in the MAVE dataset.**

| Category | Number of Nodes | | | | Number of Edges | | | Number of Hyperedges | | | | |
|---|---|---|---|---|---|---|---|---|---|---|---|---|
| | P | C | A | I | $e^{pc}$ | $e^{pa}$ | $e^{pi}$ | $\epsilon^{vp}$ | $\epsilon^{bp}$ | $\epsilon^{vi}$ | $\epsilon^{bi}$ | $\epsilon^{pi}$ |
| Industrial | 1594 | 332 | 593 | 5207 | 6611 | 2895 | 5207 | 182 | 691 | 740 | 2831 | 4612 |
| Arts | 5195 | 569 | 1023 | 19359 | 21858 | 9165 | 19359 | 851 | 1562 | 3000 | 5764 | 17532 |
| Cellphones | 7238 | 114 | 877 | 32636 | 23438 | 14905 | 32636 | 875 | 390 | 4258 | 2069 | 31880 |
| Automotive | 7768 | 534 | 863 | 24807 | 34508 | 13436 | 24807 | 895 | 47 | 3142 | 180 | 22558 |
| Office products | 7486 | 391 | 1726 | 26232 | 36456 | 15263 | 26232 | 2324 | 4139 | 11032 | 21111 | 24034 |
| Pet | 9577 | 401 | 1196 | 35407 | 43414 | 20860 | 35407 | 3524 | 1616 | 14953 | 7104 | 33096 |
| Grocery | 10652 | 666 | 2021 | 35046 | 43953 | 20890 | 35046 | 4152 | 8590 | 15881 | 33762 | 32572 |
| Tools | 20046 | 782 | 3519 | 74522 | 82277 | 42465 | 74522 | 4656 | 2421 | 21170 | 10906 | 69312 |

*Evaluation Metrics.* To better evaluate the performance of our proposed model and compare it with other baseline methods, we use the *micro-F1* score and *mAP (mean Average Precision)* as our evaluation metrics for both the main results and ablation studies. F1 score is the harmonic mean of precision and recall, and it effectively evaluates both the accuracy and completeness of predicted product attribute values, making it a suitable metric for evaluating product attribute extraction. Similarly, mAP combines both precision and recall. It also enables the performance evaluation across multiple classes of attributes and offers a comprehensive view of the model's performance. For the main results, we report the results in the form of F1/mAP (%) averaging from 10 runs. For ablation studies, we report the results in the form of F1/mAP (%) ± standard deviation.

## 5 Results

### 5.1 Main Results

We evaluate the performance of our model on eight categories of the MAVE dataset and compare it with four baseline methods. The main results are presented in Table 2. We can observe that:

(1) In terms of both F1 and mAP, our model significantly outperforms all baselines in 5 out of 8 categories. For the three categories, *Industrial*, *Cellphones*, and *Grocery*, BLIP2 and InternVL perform comparably to our model or demonstrate slight improvements in terms of F1 *or* mAP. This indicates the success and bright potential of hypergraph-based models over generative MLLMs in multi-modal multi-label zero-shot attribute value extraction.

(2) From both Table 1 and Table 2, it can be observed that larger categories, which have a greater number of nodes, edges, and hyperedges, tend to exhibit poorer performance compared to categories with fewer nodes and edges. For example, the *Tools* category, represented by larger size of graphs and hypergraph, demonstrates the lowest performance among all categories. We argue that this is due to the increased complexity and diversity of interrelations between nodes and edges/hyperedges in larger graphs or hypergraphs, making these interrelations more difficult to capture and resulting in lower model performance. We tend to explore better approaches to capture more complex interconnections among products to better support larger hypergraphs in future work.

(3) As discussed in Section 4.2, the prompt example already reveals the number of attributes and attribute name information to the baseline models. Therefore, these baseline models only need

to predict the values for the known attribute names. For example, when provided with the attribute name *Material*, the baseline models only need to select the correct attribute values from a predefined attribute value list: *acrylic, glass, pine, plastic, resin, steel, wood.* In contrast, our model does not have prior knowledge of the attribute names. It directly extracts attribute name and value pairs. For example, our model is expected to extract correct attribute-value pairs such as *"Material: acrylic", "Material: glass", and "Material: pine".* However, we never provide the potential number of attributes and attribute names for our proposed model to do the prediction. In short, all the generative MLLM baselines have both the attribute names and the number of labels as their prior knowledge. Even under these circumstances, our model consistently outperforms other baselines in most cases or demonstrates comparable performance. This highlights the model's superior ability in zero-shot attribute-value pair extraction compared to other generative baseline models.

Note that for the baseline models, we also experiment with prompts that do not explicitly point out the attribute names. We directly ask: *"What are the attribute values of this product? Choose at least one from the given choices directly: Material: acrylic, Material: glass, Material: pine, Material: plastic, Material: resin, Material: steel, Material: wood".* After extensive experiments, we find that these baseline models exhibit little or even no ability to directly extract attribute-value pairs in the zero-shot setting, performing poorly in terms of both F1 and mAP. Therefore, we choose to provide the attribute names for all generative baselines to evaluate their ability to predict attribute values in the zero-shot scenario. (Attribute names are never provided to our hypergraph-based model.)

### 5.2 Ablation Study

To evaluate the effectiveness of different components of our model, we conduct ablation studies in the following two aspects: (1) removing the visual information (i.e., product images) while retaining only the textual modality, (2) excluding hypergraphs from the framework and utilizing only graphs. The results of ablation studies are shown in Table 3. *w/o images* refers to the model without visual information, and *w/o hypergraphs* refers to the model without the hypergraph component.

**Ablating the visual information.** To evaluate the effectiveness of product images on the model's performance, we conduct a visual ablation experiment by removing the images from the input and utilizing only textual information (i.e. product titles and descriptions, attribute values, and categories). As shown in Table 3, with the

**Table 2: Results F1 / mAP (%) across eight categories in the MAVE dataset. The best performances are highlighted in bold.**

| Methods | Industrial | Arts | Cellphones | Automotive |
|---|---|---|---|---|
| Qwen-VL | 29.75 / 49.51 | 28.44 / 50.14 | 19.80 / 50.22 | 12.71 / 50.17 |
| LLaVA | 38.88 / 50.17 | 10.11 / 50.14 | 41.11 / 50.22 | 22.67 / 50.17 |
| BLIP2 | 37.70 / 49.52 | 22.55 / 50.46 | 33.00 / 49.40 | 28.65 / 49.95 |
| InternVL | **41.20** / 50.37 | 25.60 / 50.25 | 33.50 / 49.31 | 39.10 / 50.80 |
| Our model | 40.44 / **62.91** | **50.00 / 78.33** | **41.67 / 59.58** | **49.62 / 57.52** |
| **Methods** | **Office products** | **Pet** | **Grocery** | **Tools** |
| Qwen-VL | 10.91 / 50.11 | 22.90 / 50.14 | 22.37 / 50.09 | 25.11 / 50.10 |
| LLaVA | 4.77 / 50.11 | 14.92 / 50.14 | 29.98 / 50.09 | 24.47 / 50.10 |
| BLIP2 | 25.43 / 49.87 | 40.80 / 49.80 | 34.85 / 51.43 | **32.15** / 50.78 |
| InternVL | 24.42 / 50.48 | 27.60 / 50.22 | 37.80 / **51.89** | 26.50 / 51.07 |
| Our model | **59.05 / 79.17** | **60.00 / 54.33** | **40.00** / 51.70 | 26.03 /**51.28** |

**Table 3: Experimental results F1 / mAP (%) of the effectiveness of different components in our model across eight categories in the zero-shot setting. The best performances are highlighted in bold.**

| Model components | Industrial | Arts | Cellphones | Automotive |
|---|---|---|---|---|
| Full model | **40.44 ± 0.01 / 62.91 ± 0.04** | **50.00 ± 0.00 / 78.33 ± 0.04** | **41.67 ± 0.06 / 59.58 ± 0.10** | **49.62 ± 0.01** / 57.52 ± 0.07 |
| w/o images | 21.75 ± 0.01 / 37.37 ± 0.06 | 14.30 ± 0.00 / 47.49 ± 0.01 | 36.65 ± 0.02 / 67.80 ± 0.03 | 30.83 ± 0.03 / **57.57 ± 0.04** |
| w/o hypergraphs | 22.22 ± 0.00 / 14.44 ± 0.00 | 15.65 ± 0.01 / 59.40 ± 0.00 | 24.55 ± 0.01 / 53.08 ± 0.03 | 32.82 ± 0.03 / 46.61 ± 0.05 |
| **Model components** | **Office products** | **Pet** | **Grocery** | **Tools** |
| Full model | **59.05 ± 0.04 / 79.17 ± 0.04** | **60.00 ± 0.00 / 54.33 ± 0.07** | **40.00 ± 0.34 / 51.70 ± 0.00** | **26.03 ± 0.03 / 51.28 ± 0.08** |
| w/o images | 26.34 ± 0.00 / 54.19 ± 0.01 | 28.65 ± 0.01 / 46.40 ± 0.02 | 11.83 ± 0.04 / 41.66 ± 0.06 | 16.67 ± 0.00 / 40.71 ± 0.05 |
| w/o hypergraphs | 50.24 ± 0.02 / 53.45 ± 0.02 | 37.61 ± 0.08 / 33.94 ± 0.05 | 32.86 ± 0.01 / 46.47 ± 0.01 | 22.65 ± 0.07 / 16.57 ± 0.02 |

exception of a minor increase in mAP in the *Automotive* category, we observe significant decreases in both F1 and mAP across all eight categories when image input is excluded. These experimental results indicate that relying solely on textual information may lead to suboptimal model performance. The result also validates our assumptions that product images have essential semantic information and attribute values are implicitly contained within visual modality.

**Ablating the hypergraphs.** To evaluate the effectiveness of the hypergraph module within our framework, we directly employ constructed graphs without hypergraph construction. As shown in Table 3, removing the hypergraph module leads to a significant decrease in both F1 and mAP across all categories, demonstrating the effectiveness of hypergraphs in improving the model performance. We argue that this is due to the superior capacity of hypergraphs to capture more complex, higher-order interconnected relationships among different types of nodes, compared to the binary graphs. For example, the *product also view* hyperedge, captures potential interrelations among a variety of products that are likely to share similar attributes. While the binary graph, where edges connect only two nodes, cannot represent these intricate relationships among multiple products. Similarly, the *product with all images* hyperedge, which connects multiple images associated with the same product, also captures potential interrelations among images (i.e. multiple views or angles of the product) representing the same product more effectively.

### 5.3 Efficiency Study

To evaluate the efficiency of our model, we conduct efficiency studies in (1) time efficiency and (2) computational efficiency.

**Time efficiency.** Table 4 presents the inference times for our model and baseline models. From Table 4, we observe that overall, the inference time for our model on the testing set is much shorter than the time required by each baseline model. We conjecture that this is because: (1) For our hypergraph model, the inference process is not sequential. It calculates the cosine similarity between pairs of nodes in parallel for inductive link prediction. However, for generative MLLMs, it requires the model to generate the outputs (attribute values) sequentially. (2) These generative MLLM baselines have more parameters, which require more computations during inference, leading to longer processing times. (3) We can also observe that certain categories, such as Arts, Automotive, and Grocery, require longer inference times than others among baselines. We think that this is because these categories contain attributes with a large number of corresponding values, which increases the time needed to generate predictions. Remember that for generative MLLM baselines, we use a prompt to ask the model to select the correct value from all possible options. This process takes much longer time when the option set is larger. For example, the *'Type'* attribute in the *Arts* category has over 500 values, which significantly increased the time required to predict the correct values, resulting in a longer average inference time. This indicates that it is hard for generative MLLM baselines to scale up for larger categories with large quantities of possible attribute values.

**Computational efficiency.** Table 5 presents the computational efficiency of our model and baseline models. We can observe that compared to generative-based models, which exhibit high GPU memory consumption for the inference stage, our model does not require GPU utilization during inference. We argue that this is

**Table 4: Inference time (in seconds) across eight categories for our proposed model and all generative MLLM baselines.**

| Methods | Industrial | Arts | Cellphones | Automotive |
|---|---|---|---|---|
| Qwen-VL | 6.5942 | 71.5344 | 5.6962 | 111.9875 |
| LLaVA | 5.6520 | 49.3384 | 5.3908 | 74.4775 |
| BLIP2 | 4.0582 | 37.0216 | 3.9530 | 89.9124 |
| InternVL | 0.4360 | 3.0832 | 0.3932 | 9.4643 |
| Our model | 0.1527 | 0.3357 | 0.3344 | 0.3592 |
| **Methods** | **Office products** | **Pet** | **Grocery** | **Tools** |
| Qwen-VL | 5.6804 | 20.7792 | 47.4036 | 12.7758 |
| LLaVA | 4.4766 | 17.4042 | 27.9954 | 10.3977 |
| BLIP2 | 3.5157 | 12.3729 | 21.0672 | 14.3280 |
| InternVL | 0.3352 | 1.2327 | 2.0250 | 2.4048 |
| Our model | 0.4938 | 0.5746 | 1.1097 | 1.0694 |

due to our model's ability to perform zero-shot link prediction by generalizing to new product nodes and unseen attribute value nodes without full model retraining. Furthermore, in contrast to baseline models with billions of parameters, our model is more lightweight with only 117 million parameters, which contributes to its lower computational cost and the feasibility of inference running on CPUs instead of GPUs, which saves computational costs for potential deployment in e-Commerce platforms.

**Table 5: Computational efficiency during inference**

| Model | GPU Usage | Model Parameters |
|---|---|---|
| Qwen-VL | 20432MiB / 81920MiB | 9.6 Billion |
| BLIP2 | 9747MiB / 81920MiB | 3.94 Billion |
| LLaVA | 32590MiB / 81920MiB | 13 Billion |
| InternVL | 10208MiB / 81920MiB | 2.2 Billion |
| Our model | no GPU | 117 Million |

## 6  Analysis and Discussion

(1) Our model demonstrates an excellent capability for **multi-label** product attribute value extraction, allowing it to extract values for multiple types of attributes simultaneously. For example, it can extract several attribute-value pairs, such as *'Variety: peanut', 'Flavor: lightly salted', 'Preparation: roasted'*, in a single step for a given product. In contrast, baseline models are limited to single-label attribute value extraction, requiring separate prompts for each attribute type. For example, they need one prompt to extract *'peanut'* for *'Variety'* and another to extract *'lightly salted'* for *'Flavor'*. In the real-world scenario, products often possess a diverse range of attribute names and values, and extracting these individually would cost extra computational resources and increase processing and inference time. As a result, our model is capable of achieving a more efficient, comprehensive, and accurate extraction of product attributes, increasing its potential for scale-up deployment in practical applications.

(2) As illustrated in 4.2, we keep only those attributes that have six or more corresponding attribute values. The main reason for doing this is to ensure a fair comparison between our model and the baseline methods. To enable this fair comparison, we disclose the attribute name information in the prompts to the baseline models,

which then predict values for those known attribute names. For attributes with only a few corresponding values, such as one or two, the baseline models can simply select from a very limited set of attribute values without performing any actual prediction or extraction. This results in artificially inflated F1 or mAP scores. Therefore, it is necessary to exclude those attributes with very few values. After experimenting with different numbers, we choose 6 as the threshold to guarantee robust performance of the baseline models while maintaining fairness in the comparison between the baseline methods and our model.

(3) As discussed in Section 5.1, we observe that larger categories tend to demonstrate lower performance compared to those categories of smaller sizes. This is likely due to the increased complexity of interrelations within larger graphs and hypergraphs. We conduct experiments with different hyperparameter configurations and find that the performance for larger categories can be improved by increasing the batch size or training the model with more epochs. We recommend using these strategies to enhance the model performance when dealing with larger categories.

## 7  Conclusion and Future Work

In this paper, we propose a hypergraph-based model for multi-modal, multi-label product attribute value extraction in the zero-shot setting. We aim to extract previously unseen attribute values for new products, where no labeled data is available for training. Specifically, we construct multi-modal hypergraphs in which both product textual and visual information are represented as different types of nodes, enabling the model to capture semantic, complex, and higher-order correlations among product image, product text, and user behavior information. Using constructed hypergraphs, we perform zero-shot (inductive) link prediction to infer the existence of links between unseen product nodes and attribute value nodes. We conduct extensive experiments across eight primary categories on the MAVE dataset, where our model demonstrates superior performance (higher F1-score, shorter inference time, and lighter weight) over several state-of-the-art generative multi-modal large language models. For future work, we will (1) build dynamic hypergraphs using timestamps to consider the real-time market; (2) explore other approaches for constructing hyperedges.

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

## A  Baselines

Below are detailed descriptions of baseline models:

**Qwen-VL** [1] carefully designs a visual receptor, a position-aware vision-language adapter, and a three-stage training pipeline to optimize the entire model, enabling its superior visual understanding ability. We evaluate the performance of the Qwen-VL model and present the results in the following section.

**BLIP2** [26] proposes an efficient framework that pre-trains a lightweight Querying Transformer in two stages. The first stage learns a vision-and-language representation leveraging a frozen image encoder, while the second stage performs vision-to-language generative learning leveraging a frozen LLM. We conduct experiments with BLIP-2 using FLAN-T5-XL [9] as the backbone LLM model.

**LLaVA** [28] connects the pre-trained visual encoder from CLIP with an LLM decoder Vicuna [8], fine-tuning on ChatGPT/GPT-4 generated vision-language instruction-following data to enable comprehensive visual and language understanding. We evaluate the performance of LLaVA with Vicuna-13B as the backbone LLM model.

**InternVL** [7] scales up the vision foundation model to 6 billion parameters and aligns it with the LLM through three progressive stages, including vision-language contrastive learning, vision-language generative training, and supervised fine-tuning, achieving powerful visual capabilities. We conduct experiments with the InternVL2-2B variant.