# OpenReview forum: "Hypergraph-based Zero-shot Multi-modal Product Attribute Value Extraction"
_ACM.org/TheWebConf/2025/Conference — WWW 2025 Poster_

### Official Review · Reviewer_igqs · 2024-11-18

**Novelty:** 5
**Technical Quality:** 5

**Review:**

## Pros
- This paper proposes a new framework for hypergraph-based zero-shot multi-modal value extraction framework, which is an important topic in the Web community.
- The novelty of this paper is pretty good and the efficiency of the method is outperforming.

## Cons
- Some of the tables do not have a bottom line.
- Despite doing multiple experiments, the authors do not give the variance in Table2.
- More baselines should be added in the experiments. In some datasets, the method proposed in this paper seem obviously worse than the baselines.
- More case studies for both positive cases and negative cases should be conducted to discuss what is the strength of hypergraph-based model.

**Questions:**

- According to the paper, all baselines are generative MLLMs without training on the downstream datasets. However, authors train their model on the datasets and conduct the evaluation in a zero-shot setting. Is it reasonable and convincing? Nonetheless, on many datasets, your proposed method still seems to outperform the baseline. I think the main experiments should be carefully considered and revised. More baseline experiments should be conducted.

**Reviewer Confidence:**

3: The reviewer is confident but not certain that the evaluation is correct

**Scope:**

3: The work is somewhat relevant to the Web and to the track, and is of narrow interest to a sub-community

---

### Official Review · Reviewer_DTBG · 2024-11-26

**Novelty:** 5
**Technical Quality:** 5

**Review:**

Summary
To identify attribute values for the newly introduced products without similar product history records and incorporate multi-modal product information, this paper proposes a hypergraph-based zero-shot multi-modal, multi-label model that extracts multi-label product attribute-value pairs from multiple modalities, which act as different types of nodes in a hypergraph. The representations of texts and images are learned from CLIP encoders so that multi-modal product information can be incorporated and the hyperlinks between unseen product nodes and attribute-value nodes can be accurately predicted, which enables zero-shot attribute value extraction. The experimental results on the MAVE dataset manifest that the effectiveness and efficiency of the proposed model outperforms SOTA generative multi-modal LLMs in zero-shot learning.

Strengths
1. This paper identifies the challenges of the extraction of product attribute values in zero-shot learning.
2. This paper builds a hypergraph-based learning model with multi-modal information.
3. The proposed model is able to accurately predict hyperlinks with much fewer parameters than baselines (CPU only).

Weaknesses
1. The motivations can be further enhanced with more related work comparisons.
2. The details of the proposed model can be elaborated.
3. The experimental results can be further enriched.

**Questions:**

The motivations can be further enhanced with more related work comparisons.
1. The authors are encouraged to compare with studies such as [TCSS24], which focuses on multi-modal KG embedding with goals similar to extracting attribute-value pairs of products.
[TCSS24] Y. Liang, "Multimodal Knowledge Graph Embedding With Missing Data Integration," in IEEE Transactions on Computational Social Systems, 2024.
2. The authors are encouraged to compare with other hypergraph-based relation modeling using visual features such as [MM24].
[MM24] Lu Chen, Qiangchang Wang, Zhaohui Li, and Yilong Yin. 2024. Hypergraph-guided Intra- and Inter-category Relation Modeling for Fine-grained Visual Recognition. In Proceedings of the 32nd ACM International Conference on Multimedia (MM '24). Association for Computing Machinery, New York, NY, USA, 8043–8052.
3. The authors are encouraged to further consider recommendation problems based on hypergraphs, which is studied in [ICME24].
[ICME24] Y. Tan et al., "Heterogeneous Hypergraph Structure Learning for Multimedia Recommendation," 2024 IEEE International Conference on Multimedia and Expo (ICME), Niagara Falls, ON, Canada, 2024.

The details of the proposed model can be elaborated.
1. The authors are encouraged to elaborate more details of hypergraph construction, in which there are many types of hyperlinks that may make readers feel confused (especially “also view” and “also buy”).
2. It seems that the authors are focusing on link prediction in conventional graphs. Now that the hypergraph is constructed, the authors are encouraged to try more advanced objectives such as hyperlink prediction.

The experimental results can be further enriched.
1. It seems that this work is extended from HyperPAVE, which is proposed in citation [16] (also duplicated with [17]). The authors are encouraged to compare the effectiveness with HyperPAVE.
2. The ablation study may include the effectiveness with and without text data.
3. The authors are encouraged to conduct case studies to show how hyperlinks can be accurately predicted.
4. The authors are encouraged to show visualization results of the hypergraph embeddings to show how hypergraphs are represented.
5. Now that user behavioral information is exploited in the hypergraph construction, the authors are encouraged to conduct downstream tasks such as evaluating item recommendation effectiveness.

**Reviewer Confidence:**

4: The reviewer is certain that the evaluation is correct and very familiar with the relevant literature

**Scope:**

4: The work is relevant to the Web and to the track, and is of broad interest to the community

---

### Official Review · Reviewer_NNkP · 2024-12-01

**Novelty:** 5
**Technical Quality:** 4

**Review:**

The paper proposes a new graph and multi-modal language model-based method for zero-shot attribute prediction for products of e-commerce. A graph with nodes of products (descriptions), attribute values, images and categories, edges of 3 types, as well as hyper edges from user actions (e.g., buy also) is constructed, nodes features are initilized by CLIP and then learned by a GNN, and finally these GNN embeddings are used to calculate the similarities between products and new attribute values, for zero-shot prediction.

Pros:

1. The works is relevant to the Web Conference, as e-commerce is an important application of the Web and products are important Web data.

2. The method is clearly presented, with promising results achieved by the proposed method, on 8 categories of products.

Cons:

1. Important related works on knowledge graph-based (or multi-relation graph-based) zero-shot learning are ignored. Here is a comprehensive survey: https://ieeexplore.ieee.org/abstract/document/10144560.

2. The evaluation ignores the prediction over seen attribute values. Usually such zero-shot learning methods need to make a balance over the seen classes and unseen classes, and the overall performance on them should be calculated and compared.

3. The appearance of Equation (14) is a bit weird, broking the organisation of the methodology part. Before (14), 3.2.3 is introducing prediction (inference), and then it jumps to training.

**Questions:**

1. What is the performance of the proposed method and the baselines over the seen attribute values?

2. Can some methods on zero-shot knowledge graph completion or inductive knowledge graph link prediction be applied in this problem?

**Reviewer Confidence:**

3: The reviewer is confident but not certain that the evaluation is correct

**Scope:**

3: The work is somewhat relevant to the Web and to the track, and is of narrow interest to a sub-community

---

### Official Review · Reviewer_FpUc · 2024-12-02

**Novelty:** 5
**Technical Quality:** 5

**Review:**

This paper focuses on multi-modal product attribute value extraction (MAVE), and proposes a method for extracting multi-label product attribute-value pairs from multiple modalities in the zero-shot scenario. Experimental results on several categories of the public MAVE dataset demonstrate the significant advantage for multi-label, multi-modal product attribute value extraction in the zero-shot setting.

Pros:
1. MAVE task is critical for real-world scenarios.
2. The motivation is clear.
3. The proposed method is interesting.

Cons:
1. Some details are not clear. (See Questions)
2. No code and data available.

**Questions:**

1. Section 3.1 Problem Formulation needs further clarification, especially for the input and output of the task.
2. The paper turns the MAVE task into the link prediction. Compared to generative MLLMs, what is its advantage?

**Reviewer Confidence:**

3: The reviewer is confident but not certain that the evaluation is correct

**Scope:**

4: The work is relevant to the Web and to the track, and is of broad interest to the community

---

### Official Review · Reviewer_wrtW · 2024-12-04

**Novelty:** 4
**Technical Quality:** 4

**Review:**

The paper proposes a hypergraph-based zero-shot multi-modal product attribute value extraction framework for e-commerce platforms. Its key contributions include the use of a heterogeneous hypergraph to model complex relationships among multiple modalities as well as user behavior data. The method leverages CLIP encoders to extract multi-modal representations and performs zero-shot link prediction to infer attribute-value pairs for unseen products. Extensive experiments on the public MAVE dataset demonstrate that the proposed method outperforms SOTA MLLMs in terms of both prediction performance and computational efficiency.

Pros:
- The proposed method demonstrates strong zero-shot learning capabilities, making it applicable to real-world scenarios where labeled data is scarce.
- The proposed method addresses a significant real-world challenge in e-commerce by enabling accurate and efficient extraction of multiple attribute-value pairs, reducing manual effort and improving product search and recommendation systems.

Cons:
- Scalability concerns. While the method is computationally efficient compared to SOTA baselines, the scalability of the hypergraph construction and message-passing mechanisms for very large datasets is not discussed in detail. Specifically, as you increase the number of nodes, how would the inference time rises?

**Questions:**

- While the method is computationally efficient compared to SOTA baselines, the scalability of the hypergraph construction and message-passing mechanisms for very large datasets is not discussed in detail. Specifically, as you increase the number of nodes, how would the inference time rises?
- How would the proposed model compare with GPT-4o? If GPT-4o is better, what are the unique advantages of the proposed method?

**Reviewer Confidence:**

2: The reviewer is willing to defend the evaluation, but it is likely that the reviewer did not understand parts of the paper

**Scope:**

2: The connection to the Web is incidental, e.g., use of Web data or API